# Assessment of Interannual Variability of Moistening of Siberian Territory According to Observational Data

**Valeriy Malinin [1], Svetlana Gordeeva [1,2,*] and Julia Mitina [1]**

[1] Institute of Hydrology and Oceanology, Russian State Hydrometeorological University, 192007 Saint-Petersburg, Russia; malinin@rshu.ru (V.M.); mitinarshu@mail.ru (J.M.)

[2] The St.-Petersburg Department, Shirshov Institute of Oceanology of Russian Academy of Sciences, 199053 Saint-Petersburg, Russia

[*] Correspondence: gordeeva@rshu.ru; Tel.: +7-911-822-2920

**Abstract:** The article discusses the features of large-scale spatial and temporal variability of moistening (potential evapotranspiration, precipitation, potential evapotranspiration coefficient) in the Siberian part of Russia for the period 1981–2015. The All-Russian Research Institute of Hydrometeorological Information—World Data Center (RIHMI-WDC) archive has served as a source of initial information. Due to the rare network of stationary meteorological stations in most of Siberia, only 32 stations located mainly in the valleys of large rivers have been used for calculations. To estimate potential evapotranspiration, the modified method of M.I. Budyko has been used. A comprehensive delimitation of Siberia has been carried out by the interannual fluctuations of characteristics of moistening, being well divided into four regions, three of which encompass the basins of the largest rivers: the Ob, the Yenisei, the Lena and the fourth region represents the Baikal region. Analysis of the trends shows that the evapotranspiration in Siberia is growing only in the Ob basin and the Baikal region. Precipitation, excluding the Baikal region, is also increasing in the Yenisei and Lena basins. As for the potential evapotranspiration coefficient, a significant trend refers only to the Baikal region due to the rapid increase in evaporation. The modeling of the annual values of the characteristics of moistening for the selected regions has been carried out using the decision trees method. For 4-branch trees, the coefficient of determination $R^2$ describes about two-thirds of the variance of the original variable (0.57–0.73). In the models of annual evapotranspiration values, the main predictor is the air temperature. In precipitation models, the contribution of local and external circulation factors to interannual precipitation fluctuations is equal.

**Keywords:** moistening; Siberia; potential evapotranspiration; precipitation; potential evapotranspiration coefficient; trend; decision trees

## 1. Introduction

Absolute moistening is usually understood as the difference between the amount of precipitation ($P$) and evaporation from the underlying surface ($E$). Moistening forms the inland waters, surface and ground-water storage and affects the state of biocenoses. If $P > E$ for a long time, then excessive moistening is noted; if $E > P$, then it leads to depleting of soil-water storage and, consequently, to droughts.

Due to rapid acceleration of global warming, there is an increase in dangerous hydrometeorological phenomena, which include, among other things, droughts (arid moistening) and floods (excessive moistening). At the same time, the differentiation in moistening increases, i.e., the arid regions, as a rule, become even more arid, with river floods intensifying [1–3]. For example, over the past 10 years, the strongest floods ever recorded happened twice on the Amur River (in 2013 and 2019). It is the large-scale droughts and floods that lead to significant environmental and economic damage. Therefore, identifying the genesis of interannual variability, and constructing models for long-term forecast of characteristics

of moistening under the conditions of modern climate change is not only a fundamental scientific problem, but also is of great ecological and economic importance.

When assessing the difference $P-E$, the greatest difficulty is caused by assessment of total evaporation, especially from large areas consisting of a combination of different types of land surface, such as forests, meadows, agricultural fields, water bodies, stream channels, swamps, urbanized areas, etc. Evaporation is possible only when the equation of water balance of a river basin or the atmosphere is closed. As a result, the estimate of total evapotranspiration will include all errors of other balance components. Accounting for individual types of evaporation being complex, semi-empirical relationships between total evaporation and its determining factors are often used in calculations. For this purpose, the value of potential evapotranspiration ($E_0$) is most often used, being normally understood as the maximum possible evaporation under given meteorological conditions, when soil moisture has no limiting effect on the evaporation process. This value gives an idea of the upper limit of evaporation from land when there is no lack of moisture in the soil [4].

Potential evapotranspiration is the basis for assessing different indices of moistening: potential evapotranspiration coefficient $E_0/P$, relative potential evapotranspiration $E/E_0$, potential evapotranspiration deficit $E_0-E$, etc. [5,6]. The most universal index is the potential evapotranspiration coefficient $E_0/P$, which is directly related to $P-E$, total evaporation $E$ and other characteristics of moistening through various semi-empirical equations. For example, in general, the equation between the water balances of the atmosphere and the land surface is [7]:

$$(E - P)/P = \varphi\ (E_0/P),$$

where $\varphi$ is a function dependent on a geobotanical zone. This equation has three following features:

(1)  it reflects the interrelation of parameters that cause moistening;
(2)  the difference $E-P$ enters simultaneously into the equations of balance of the atmospheric and terrestrial branches of the hydrological cycle, thus linking the transport of moisture in the atmosphere with the one of water in the soil-ground layer;
(3)  it is carried out most accurately at sufficiently large spatiotemporal averaging scales, when the total evaporation is determined mainly by climatic factors, and the role of local (landscape) factors can be neglected. The specific formula for $\varphi$ is given in [7,8].

The purpose of this paper is to identify the features of the spatiotemporal variability of moistening and delimitate the Siberian part of Russia. There are a number of works that discuss various aspects of the climate of Siberia [1,9–14], the most detailed comprehensive description of variability of the modern climate of Siberia being presented in the monograph [1]. Due to its huge area, Siberia has been divided into 5 regions: Western, Central, Eastern Siberia, Baikal zone, Priamur'ye and Primor'ye (Figure 1). Significant positive trends in the annual air temperature and annual rainfall are shown to be present for the period 1976–2012 for the entire territory of Siberia, excluding Eastern Siberia, where precipitation decreases over time [1].

A detailed analysis of intra-annual and interannual cyclonic activity and its relationship with precipitation over different time periods for the Ob, Yenisei and Lena basins is given in the detailed monograph [9]. The average long-term distribution of the characteristics of moistening over the plains of Western Siberia is considered in works [10,11]. Maps of average long-term monthly values of various parameters of moistening (potential evapotranspiration, potential evapotranspiration coefficient $E_0/P$, relative evaporation $E/E_0$) over the territory of the former USSR are given in [5] and characterize climatic conditions of moistening prior to 1970. For most of Siberia, these values are very approximate due to poor coverage by hydrometeorological stations, complex terrain, and significant spatial heterogeneity of meteorological characteristics. The interannual variability of moistening in Siberia has scarcely been studied.

This work is the first to present a comprehensive zoning of the territory of Siberia according to the nature of interannual fluctuations in the annual values of characteristics of

moistening, and to identify four quasi-homogeneous regions along with modeling their annual values of characteristics of moistening by the decision trees method using the CART algorithm.

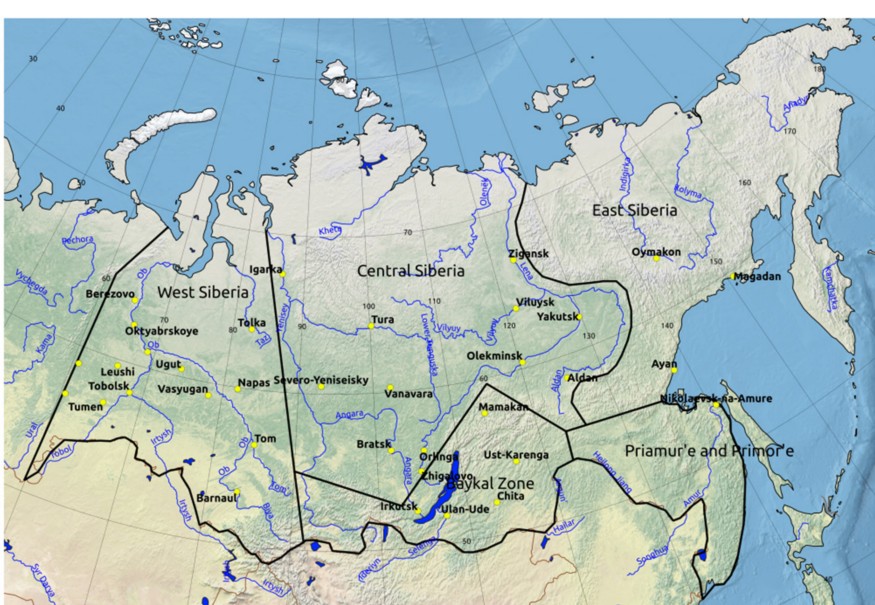

**Figure 1.** Map of Siberian territory. Yellow circles are meteorological stations. Black lines are boundaries of Siberian districts (approximately, according to [1]).

## 2. Materials and Methods

The archive of All-Russian Research Institute of Hydrometeorological Information–World Data Center (RIHMI-WDC) (http://aisori.meteo.ru/ClimateR, accessed on 1 June 2021), containing data of standard instrumental observations at almost 500 meteorological stations in Russia, predominantly since 1950, served as the main source of the initial meteorological information. In this work, we used data on air temperature, relative humidity, and precipitation. This archive provides the precipitation time series starting in 1966, when the Tretyakov rain-gauge was completely introduced at meteorological stations; since January 1966, a wetting correction has been introduced. There have been no changes in the measuring and processing techniques since 1966; therefore, the series of rainfall can be considered homogeneous [15]. However, as precipitation values at the stations are corrected only for wetting, we performed an additional correction of all systematic errors of annual rainfall in accordance with the methodology of [16], resulting in their increase by 8–20%.

The catchment areas of three largest rivers (the Ob, the Yenisei, the Lena) were taken as the territory of Siberia. A serious problem was the extremely low density of the network of stations, which are located mainly in the valleys of large rivers, thereby only approximately reflecting the conditions of moistening throughout the vast territories between them. Even in the southern regions, where most of the population of Siberia lives, the distances between the stations are considerable. As a result, only 32 stations could be selected without data gaps. Figure 1 shows a map of the territory of Siberia with the plotted stations used in the work, which are practically all located in the geobotanical zone of the taiga. We should note that the territory of Priamur'ye and Primor'ye was disregarded due to the fundamental difference in its climatic conditions when compared to other regions of Siberia. It should be also stated that the area of Siberia considered in this work is approximately 10 million square kilometres.

The main parameter of moistening is potential evapotranspiration. A considerable number of methods are known for its evaluation [4,10,17–26], each of them having its own advantages and disadvantages. Comparison of some of these methods for different territories does not show an obvious advantage of any one method for assessing evapotran-

spiration over others. This, for example, is evidenced by works [27–30], which compare six different methods for the territory of Poland, the southeast and southwest of the USA, Japan and central Europe. Work [31] presents the maps of long-time average annual potential evapotranspiration for European Russia based on 13 methods of its calculation. The discrepancies in the $E_0$ estimates are quite significant, but according to the authors, the Thornthwaite method [23] is preferable due to its "using only air temperature data". However, this can hardly be called a serious reason, since there are many empirical formulas based solely on air-temperature data. The Thornthwaite's formula is certain to primarily reflect the climatic conditions of the United States. Significant errors in the estimates of potential evapotranspiration by the Thornthwaite method, especially in winter and mid-seasons, are noted in [5].

In recent years, the Penman-Monteith model [32–34], recommended by experts from the Food and Agriculture Organization of the United Nations, appears to be used most often in studies to assess evapotranspiration. Although the method is physically justified, it requires knowledge of a number of difficult to determine parameters associated with the process of heat transfer in the soil and is tied to specific types of soils [35]. From our point of view, when applied to the territory of Russia, the best adapted method is the complex method of M.I. Budyko [4], which is also physically justified and can be used for any kind of underlying surface. Despite its rather considerable "age", this method is successfully applied in modern conditions [36–38], etc.). Moreover, in work [39], Budyko's equation is said to have "achieved iconic status in hydrology for its concise and accurate representation".

The Budyko method is based on the idea of proportionality of evaporation from a wet land surface to a moisture deficit, determined by the temperature of evaporating surface. A comparative analysis of individual element contribution to the $E_0$ estimate shows that the effect of cloudiness, used to determine the radiation balance, can be neglected, temperature and humidity being related through moisture deficit $d$. Therefore, $E_0$ can be expressed in terms of moisture deficit.

The main disadvantage of the method is that it can be used to calculate only long-term monthly average values of $E_0$. To determine $E_0$ for specific monthly time intervals, one can use the hypothesis of conjugation of space-time fluctuations in potential evapotranspiration and moisture deficit [6,8], according to which the evaporation in a specific $i$-th month at the $j$-th station is determined as:

$$E_{0ij} = E_{0j} \, (d_{ij}/d_j)^{\chi}, \tag{1}$$

where $E_{0j}$ and $\overline{d}_j$ are, respectively, the long-term average annual values of potential evapotranspiration and moisture deficit at $j$-th station. At the same time, the former can be expressed as:

$$E_{0j} = \overline{E}_0 > (d_j \, / < d >)^{\chi}, \tag{2}$$

where the triangular brackets denote spatial averaging, the parameter $\chi$ being the ratio of the spatial (temporal) coefficients of variation ($C$) of potential evapotranspiration and moisture deficit, i.e.,

$$\chi = C_{<Eo>}/C_{<d>} = C_{\,Eoi}/C_{di/} \tag{3}$$

This formula shows that the temporal variability of monthly average parameters of potential evapotranspiration and moisture deficit at a particular point is equivalent to spatial variability of long-term average values of the same parameters. So, Formula (2) represents the desired solution to the problem of determining the potential evapotranspiration for short (monthly) time intervals. It should be noted that the proposed approach is very simple and does not require knowledge of any parameters that are difficult to determine.

For the calculations to be convenient, the parameterization of long-term average values of potential evapotranspiration was carried out. For this, we used the nomograms of their dependence on moisture deficit for various geobotanical zones of Russia, constructed for each month of the warm period (April-October) based on the integrated method and

presented in work [40]. The numerical values of the potential evapotranspiration obtained in this way were used to approximate the dependence $E_0 = f(d)$ using various empirical formulas. The most universal function proved to be the rational one [41], providing the most adequate accuracy of the potential evapotranspiration and having the following form:

$$E_0 = (a_0 + a_1 \, \overline{d})/(1 + a_2 \, \overline{d} + a_3 \, \overline{d}^2). \tag{4}$$

For each month of the warm period (April–October) and various geobotanical zones, the rational function coefficients were determined, the coefficient of determination reaching $R^2 = 0.99$ in all cases. Since the potential evapotranspiration in the winter period is taken to be equal to the evaporation from the snow, the approximate formula can be used to estimate the average monthly values:

$$E_{0 \, win} = 0.3 n \times d, \tag{5}$$

where $n$ is the number of days in a month. As a result, the annual cycle of the potential evapotranspiration is determined only from the data on the moisture deficit values. This made it possible in [41] using the Formula (3) to calculate seasonal variation of the $\chi$ coefficients for various geobotanical zones of Russia in the period 1966–1995.

The $E_0/P$ ratio values were calculated using data on water vapour pressure, air temperature and precipitation at 32 meteorological stations located mainly in the valleys of large rivers in Siberia and being assigned to the geobotanical zone of the taiga (coniferous forests). The northern part of the region is included in the tundra zone; therefore, it is not considered here. The monthly average values of the moisture deficit $d$ were calculated from the data on air temperature and relative humidity independently for positive and sub-zero air temperatures using psychometric tables.

Due to the huge meridional extent of Siberia, climatic conditions significantly differ even in the north and south of the taiga zone [1]. Firstly, this concerns the duration of the cold season, when the potential evapotranspiration is determined by the Formula (5). To separate the warm and cold periods more accurately, we used information on the monthly snow-cover thickness on a grid cell in the vicinity of the station, taken from the ECMWF ERA5 reanalysis (https://cds.climate.copernicus.eu/cdsapp#!/dataset/10.24381/cds.68d2 bb30?tab=overview, accessed on 1 June 2021). If the thickness was less than 0.1 cm, the month was assigned to the warm period. The duration of winter is 10 months and 8 months for the northern and southern regions of Siberia, respectively. For the warm period, the long-term monthly average values of $E_0$ were first calculated by the Formula (4), and then used to calculate the potential evapotranspiration using Formula (1) for specific monthly time periods at specific stations. Summing up the average monthly estimates of $E_0$, it is easy to obtain the annual amounts of $E_0$ and calculate the values of $E_0/P_{\mathrm{year}}$. It should be noted that due to the use of the ECMWF ERA5 reanalysis archive, the length of the time series was 35 years (1981–2015).

As a result of calculations, $32 \times 35$-sized matrices of annual values of the potential evapotranspiration, precipitation and coefficients of moistening were obtained. These matrices were used to delimitate the territory of Siberia by the nature of interannual variability using a hierarchical procedure of cluster analysis [42]. We used an analogue to the Euclidean metric $d' = 1 - R$, where $R$ is the correlation coefficient between the variables. The optimal number of classes was found to be 4 for all three regionalization.

Characteristics of moistening averaged for each region were used to build low-parameter (concise) models using the decision trees method, being a part of the multidisciplinary Data Mining system [43,44]. A description of the method can be found in [45,46]. Its doubtless advantage is in providing visualization of the results obtained and their physically grounded interpretation. Among the most popular tree method algorithms is the CART (Classification and Regression Tree) algorithm, developed by four professors of statistics from leading American universities [46] and solving classification and regression problems [47–49].

However, this method is still not widely used in hydrometeorological research. Nevertheless, several works [50–54] are known to use it to solve various problems. For example, in [50,51], a forecast of the annual runoff of large rivers in the European Russia (the Volga, the Severnaya Dvina, the Neva) is made, based on precipitation data for the previous year using a limited number of meteorological stations. The decision trees method is shown to provide a long-term forecast of the annual runoff with an accuracy sufficient for practical purposes. At the same time, comparison with the classical method of multiple regression analysis demonstrates its higher efficiency.

Modeling of the annual values of potential evapotranspiration and precipitation for four quasi-homogeneous regions of Siberia was carried out using the CART algorithm with a priori probabilities proportional to the number of classes, the cost of classification error being the same for all classes (Interactive Trees). In this case, minimizing losses is equivalent to minimizing the proportion of incorrectly classified observations. It should be noted that since there is usually no need to construct a complete tree, there arises a rather difficult and controversial question of stopping the tree branching. In this work, in order to compare the obtained simulation results, a variant of a 4-branch tree was adopted as the optimal model.

Statistical assessment of accuracy of results of reproducing each annual value of evapotranspiration or precipitation by the CART algorithm was carried out on the basis of traditional estimates of quality of long-term forecasts: if a standard error of a model for the entire series is less than a standard deviation of the initial time series, then a model satisfactorily describes the initial data.

Since the potential evapotranspiration depends mainly on local hydrometeorological factors, air temperature ($Ta$) and atmospheric pressure ($Pa$) were taken as the predictors for the construction of decision trees models at four stations in each region (Oktyabrskoe, Irkutsk, Severo–Yeniseisky, Oymyakon). A much more complicated task is to select predictors for precipitation, which depends on a number of external circulation factors and local conditions, namely $Pa$ in Oymyakon and Oktyabrsky. The $Pa$ in Oymyakon can be considered as a power indicator of the Siberian pressure maximum. The higher it is, the lower is the probability of precipitation over the entire area of Siberia. The $Pa$ in Oktyabrskoye characterizes the local precipitation in western Siberia and partly in central Siberia. The Oymyakon–Oktyabrskoe pressure gradient ($\Delta Pa$) characterizes the intensity of the meridional circulation, which can affect the cyclonic activity of the atmosphere.

We also used external circulation factors (https://www.cpc.ncep.noaa.gov/data/teledoc/telecontents.shtml, accessed on 1 June 2021) [55,56]:

− EAWR (East Atlantic–Western Russia)—an index characterizing four centres of atmospheric pressure: over the East Atlantic, Europe, Central Russia (eastern Siberia) and China;
− NAO (North Atlantic Oscillation) is the difference in atmospheric pressure between the Azores maximum and the Icelandic minimum and reflects the intensity of geostrophic western circulation in Europe and most of Siberia;
− SCAND reflects the presence of blocking anticyclones over Scandinavia and Northwest Russia;
− Polar/Eur characterizes fluctuations in the intensity of the circumpolar circulation, the positive and negative phases reflecting an enhanced circumpolar vortex and a weaker polar vortex, respectively;
− AO (Arctic Oscillation) reflects oppositely directed pressure changes in the Arctic and temperate latitudes and is defined as the first mode of decomposition into the main components of pressure at sea level in the Northern Hemisphere (20–90° N)
− PNA (Pacific/North American Pattern) is an index that represents the pressure difference between the Aleutian minimum and the high-pressure area over the Rocky Mountains in America. The index is associated with fluctuations in the intensity and location of the East Asian jet stream.

We should note that Marshall [57] used similar climatic indices of the North Atlantic and the northern Pacific influencing air-masses transfer (NAO, EAWR, SCAND, POL, PNA,

etc.) to describe the winter air temperature and precipitation in Siberia north of 60° N taken from ERA5 and JRA-55 archives. This allowed him to qualitatively assess the relative role of each index in the variability of temperature and precipitation in northern Siberia.

## 3. Results and Discussion

Table 1 shows the statistical parameters of the long-term average values of potential evapotranspiration and moisture deficit for the Siberian stations. It is easily seen that the summer season manifests itself only in July and August at all stations of Siberia. In September, the summer season is observed only at the southernmost stations. The calculated values of variation coefficients of potential evapotranspiration and moisture deficit allowed us to determine the coefficient χ required to calculate the monthly average values of potential evapotranspiration. Their comparison with similar estimates for the coniferous forest zone in the European Russia [41] showed their close agreement.

**Table 1.** Distribution of statistical parameters of potential evapotranspiration components for Siberian stations for 1981–2015.

| Parameter | June | July | August | September |
|---|---|---|---|---|
| Number of stations | 25 | 35 | 35 | 5 |
| Average potential evapotranspiration, mm | 116.47 | 112.73 | 81.96 | 57.55 |
| Variation coefficient of potential evapotranspiration | 0.08 | 0.12 | 0.12 | 0.18 |
| Average moisture deficit, mbar | 6.09 | 5.90 | 3.95 | 2.59 |
| Variation coefficient of moisture deficit | 0.25 | 0.26 | 0.23 | 0.25 |
| Coefficient $\chi = C_{<Eo>}/C_{<d>}$ | 0.33 | 0.45 | 0.54 | 0.72 |

Naturally, most of Siberia is located in the zone of excessive moistening since $P > E_0$. Insufficient moistening is observed mainly in Transbaikalia, in the southern Buryatia and the Chita region. The average long-term potential evapotranspiration over the flat land, primarily in Western Siberia, increases rather uniformly from north to south due to the law of latitudinal zonality—in the northern regions averaging to 350–380 mm/year. Its maximum values are observed in Transbaikalia, in the Ulan-Ude region, with the potential evapotranspiration reaching 580 mm/year. For the flat land, the dominant effect of air temperature on evaporation is quite clearly seen. However, the mountainous relief of most of Siberia has a significant influence on the distribution of potential evapotranspiration. The author [5] notes that it decreases, on average, with a gradient of 1–3 mm for every 100 m of rise, depending on the moistening of the area. Such a decrease can be significant and amount to 30–60 mm. These estimates were made in 1976; however, there are no more recent estimates. Obviously, the construction of maps of evapotranspiration and other characteristics of moistening for the territory of Siberia having extremely heterogeneous topography based on data from only 32 stations is impractical, since this creates huge uncertainties that are very difficult to quantify and even more so to control.

The terrain has a very strong influence on the spatial distribution of precipitation as well. Even in relatively small areas, the amount of precipitation can differ by 2 times. Nevertheless, the main character of precipitation in the West Siberian Plain is zonal and characterized by a general decrease in precipitation southwards [58], with an increase in precipitation in the foothill regions. The development of the Eastern Siberian climate is determined by its great remoteness from the sources of moistening in the Atlantic and presence of the mountain ranges acting as a barrier to the Pacific Ocean. Here, the general trend of decreasing precipitation from west to east persists with the allocation of considerably dry areas.

At the next stage, the delimitation of the territory of Siberia was carried out according to the nature of interannual fluctuations in the annual values of potential evapotranspi-

ration, precipitation, and potential evapotranspiration coefficient. All three delimitations were found to be close to each other, discrepancies being observed mainly for some border stations. Therefore, a comprehensive delimitation was carried out for all characteristics of moistening, the results of which are presented in Figure 2. It is easily seen that four regions within which the stations are grouped can be identified in the territory of Siberia. Three of them are the basins of the largest rivers: the Ob, the Yenisei, and the Lena, with another 3 stations located nearby the Sea of Okhotsk being roughly assigned to the Lena region. The fourth region—Baikalsky—represents the area around Baikal Lake. The results indicate that the differences in the interannual variability of moistening between river catchments are noticeably stronger than within each of them. It should be noted that the boundaries between the regions have not been drawn due to the small number of stations. Within the delimitated regions, the correlation of the annual values of each parameter ($E_0$, $P$ and $E_0/P$) between the stations is higher than with the stations of neighbouring regions.

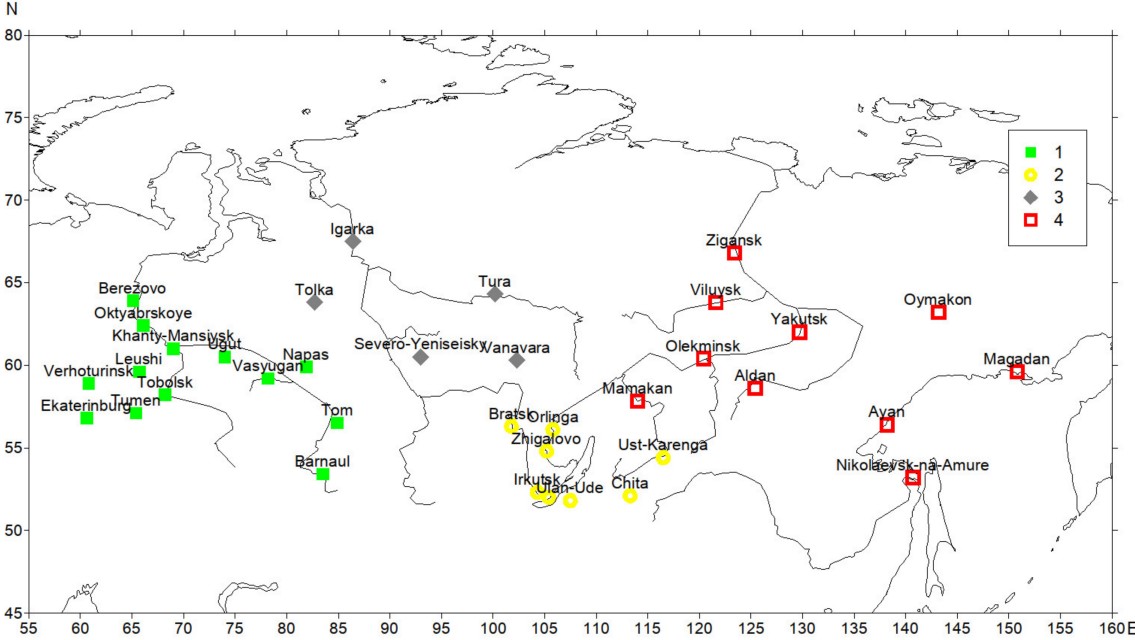

**Figure 2.** Delimitation of the territory of Siberia into 4 regions (different colours) by interannual variability of potential evapotranspiration, precipitation, and potential evapotranspiration coefficient for 1981–2015.

Table 2 gives the distribution of statistical characteristics for the annual values of the components of moistening, averaged over the selected areas for 1981–2015. As expected, the highest potential evapotranspiration is characteristic of the southernmost Baikal region. Its lowest values are observed in the Yenisei basin.

The largest amount of precipitation falls in the Ob basin. A rather strong influence of the Atlantic is still being manifested here. A significant amount of precipitation falls from local cyclones. The share of precipitation from West Siberian cyclones is 40%, from European cyclones, 35%; the rest of the precipitation falling from cyclones that come from the north and south [9,59]. In the Lena basin, the contribution of precipitation from the local area of cyclone generation is over 80% [9].

The Baikal region has the least precipitation, particularly in the southern Buryatia and the Chita region, where a continental type of climate affected by the Siberian anticyclone dominates. A series of mountain ranges prevents the passage of cyclones here. The main part of atmospheric precipitation falls in the summer season from air masses of Atlantic origin [9].

**Table 2.** Distribution of statistical parameters of potential evapotranspiration components for Siberian stations for 1981–2015. Bold values mean significant trends (by Student's test at $\alpha = 0.05$).

| Parameter | Mean | RMSD | C | Linear Trend | |
| --- | --- | --- | --- | --- | --- |
| | | | | Coefficient | $R^2$ |
| The Ob's Basin | | | | | |
| $E_0$, mm | 444.01 | 34.38 | 0.08 | 0.97 mm/year | **0.11** |
| $P$, mm | 601.88 | 64.63 | 0.11 | 1.63 mm/year | 0.07 |
| $E_0/P$ | 0.75 | 0.13 | 0.17 | $-0.0001$ year$^{-1}$ | 0.00 |
| The Yenisei's Basin | | | | | |
| $E_0$, mm | 376.40 | 27.61 | 0.07 | 0.67 mm/year | 0.06 |
| $P$, mm | 442.63 | 56.11 | 0.13 | 2.40 mm/year | **0.19** |
| $E_0/P$ | 0.73 | 0.14 | 0.19 | $-0.004$ year$^{-1}$ | 0.08 |
| The Lena's Basin | | | | | |
| $E_0$, mm | 407.96 | 28.11 | 0.07 | 0.35 mm/year | 0.02 |
| $P$, mm | 462.34 | 53.33 | 0.12 | 1.98 mm/year | **0.16** |
| $E_0/P$ | 1.01 | 0.16 | 0.16 | $-0.004$ year$^{-1}$ | 0.06 |
| The Baikal Region | | | | | |
| $E_0$, mm | 472.11 | 32.31 | 0.07 | 1.89 mm/year | **0.39** |
| $P$, mm | 374.28 | 58.72 | 0.16 | $-1.26$ mm/year | 0.05 |
| $E_0/P$ | 1.47 | 0.34 | 0.23 | 0.015 year$^{-1}$ | **0.21** |

The interannual variability of $E_0$ is almost equal in all 4 regions. The coefficient of variation is 7–8%. The interannual variability of precipitation is significantly higher, the coefficient of variation ranging from 11% (the Ob basin) to 16% (the Baikal region). The variability of $E_0/P$ is even higher, reaching 23% in the Baikal region.

Table 2 also shows the parameters of linear trends: the trend coefficient and the coefficient of determination, defining the contribution of trend to the variance of the original series. Significant trends (by Student's test at $\alpha = 0.05$) are highlighted in bold. The potential evapotranspiration in Siberia is growing, but the trend is significant only in the Ob basin and the Baikal region. Precipitation, excluding the Baikal region, is increasing as well. Significant trends are observed in the Yenisei and Lena basins. As for the potential evapotranspiration coefficient, a significant trend takes place only in the Baikal region due to the rapid growth of $E_0$.

Table 3 shows the results of modeling the annual values of potential evapotranspiration in the period 1981–2015 using the CART algorithm for four regions of Siberia.

Each model mainly has 4 branches and, accordingly, 4 (or 5) predictors, given in Table 3 in order of input to the model. The first predictor, as expected, is the air temperature at the station in the area under consideration. The coefficient of determination $R^2$ is approximately the same and describes about two-thirds of the variance of the original variable (0.59–0.64). At the same time, the Ob region, notable for its most homogeneous natural landscape, has the highest accuracy. RMSE normalized by the standard deviation (NRMSE) is significantly lower than unity, which also indicates a fairly high accuracy of the results.

**Table 3.** Estimates of modeling the annual values of potential evapotranspiration for four quasi-homogeneous regions of Siberia using the CART algorithm for 1981–2015.

| Region | Number of Branches | Predictors by Their Contribution to $R^2$ | $R^2$ | NRMSE |
|---|---|---|---|---|
| The Ob's basin | 4 | *Ta* Oktyabrskoye<br>*Pa* Oktyabrskoye<br>*Ta* Severo-Yeniseisky<br>*Ta* Oktyabrskoye | 0.64 | 0.59 |
| The Yenisei's basin | 4 | *Ta* Severo-Yeniseisky<br>*Pa* Oimyakon<br>*Pa* Oimyakon<br>*Ta* Severo-Yeniseisky | 0.59 | 0.63 |
| The Lena's basin | 5 | *Ta* Severo-Yeniseisky<br>*Ta* Oimyakon<br>*Ta* Irkutsk<br>*Ta* Oimyakon<br>*Pa* Oimyakon | 0.59 | 0.69 |
| The Baikal region | 4 | *Ta* Irkutsk<br>*Ta* Irkutsk<br>*Pa* Oktyabrskoye<br>*Ta* Severo-Yeniseisky | 0.61 | 0.62 |

Figure 3 shows a decision tree of 4 branches for the annual values of the potential evapotranspiration of the Ob region. As it can be seen from Figure 3, the first separator is the air temperature in Oktyabrskoye settlement. The initial sample is divided into parts (21 and 14 values) with air temperature being $-1.36\ ^{\circ}$C. The average $E_0$ value is 428 mm/year for a sample of 21 values and 468 mm/year for a sample of 14 values. Further, the sample of 21 values is divided into 2 unequal parts (1 and 20 values) when the air temperature in Severo–Yeniseisky is equal to $-5.53\ ^{\circ}$C, with an extremely high value of $E_0$ = 485 mm/year being separated. The third separator is atmospheric pressure in Oktyabrskoye settlement. With $Pa$ = 1013.2, the sample of 14 values is halved with the average values of 446.7 and 489.2 mm/year. Finally, air temperature in the Oktyabrskoye settlement again participates in the 4th branching. With air temperature $-3.04\ ^{\circ}$C, the sample is divided into 4 and 16 values.

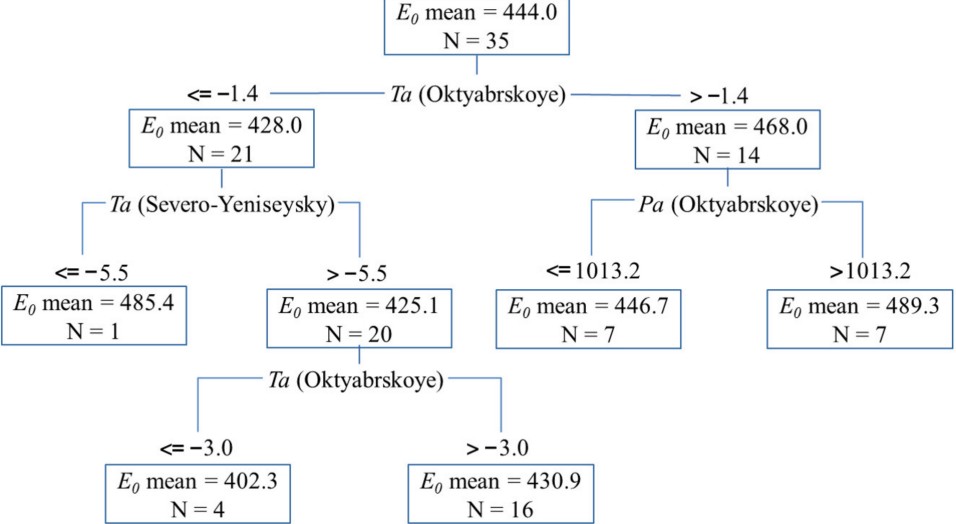

**Figure 3.** Decision tree for annual values of potential evapotranspiration (mm/year) of the Ob's basin, consisting of 4 branches for 1981–2015.

A comparison of the annual values of potential evapotranspiration calculated using this model and its actual values for the Ob region is shown in Figure 4. Their agreement is easily seen, the maximum discrepancy (1997) being 40 mm/year, or 9%.

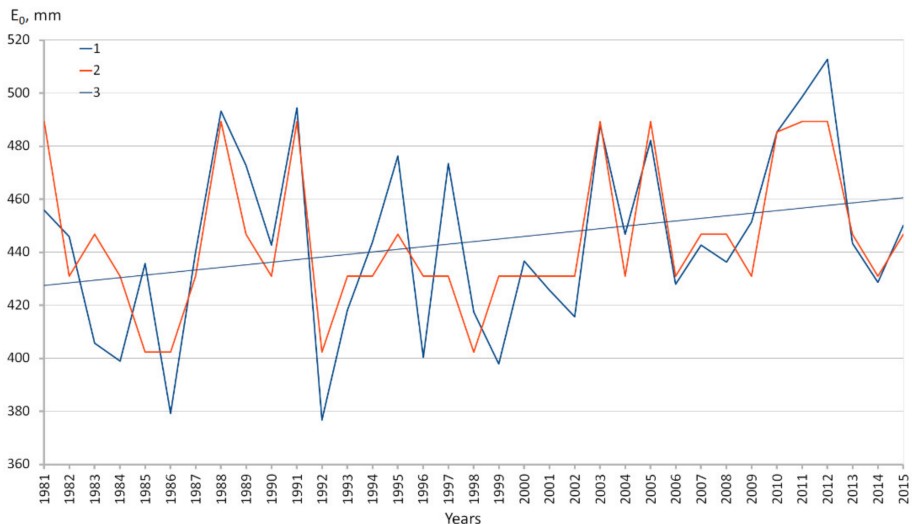

**Figure 4.** Comparison of the actual (1) (with linear trend (3)) and calculated by the decision trees model (2) annual values of potential evapotranspiration of the Ob's basin for 1981–2015.

Table 4 presents the results of modeling the annual precipitation values in four regions of Siberia in the period 1981–2015. As in the previous case, each model consists of 4 branches and, accordingly, 4 predictors. At the same time, the contribution of local (8) and external (8) factors to interannual fluctuations in precipitation is equal. Somewhat unexpectedly, the coefficient of determination in precipitation models (0.73–0.57) turned out to be higher than in evapotranspiration models. The interannual variability of precipitation in the Yenisei region is described worst ($R^2 = 0.57$). Naturally, the standard error of the precipitation estimate is also slightly less than that of evapotranspiration one.

**Table 4.** Estimates of modeling the annual values of precipitation for four quasi-homogeneous regions of Siberia using the CART algorithm for 1981–2015.

| Region | Number of Branches | Predictors by Their Contribution to $R^2$ | $R^2$ | NRMSE |
|---|---|---|---|---|
| The Ob's basin | 4 | *Pa* Oktyabrskoye<br>PNA<br>NAO<br>*Pa* Oktyabrskoye | 0.61 | 0.62 |
| The Yenisei's basin | 4 | EAWR<br>NAO<br>*Pa* Oimyakon<br>PNA | 0.57 | 0.54 |
| The Lena's basin | 4 | Δ *Pa* Oimyakon- Oktyabrskoye<br>Δ *Pa* Oimyakon- Oktyabrskoye<br>*Pa* Oimyakon<br>NAO | 0.73 | 0.45 |
| The Baikal region | 4 | SCAND<br>*Pa* Oimyakon<br>Δ *Pa* Oimyakon- Oktyabrskoye<br>NAO | 0.65 | 0.54 |

Figure 5 shows a decision tree of 4 branches for the annual precipitation values of the Ob region. At the first branching, the separator is the atmospheric pressure in the

Oktyabrskoye settlement. If it is less than 1011.7 hPa, abnormally high precipitation with an average value of 691 mm/year is observed in 5 cases out of 35. The pressure being higher than 1011.7 hPa, precipitation falls slightly less than the norm (averaged 587 mm/year) in 30 cases out of 35. The second branch is depending on the PNA index. With its large positive values (PNA > 0.51), there are 3 cases of abnormally low precipitation (averaged 515 mm/year). From the remainder 27 cases, the NAO index has an extremely low amount of precipitation (494 mm/year). Finally, at the fourth branching, the group of 26 cases is divided into precipitation slightly above normal (average 610 mm/year) and below normal (average 560 mm/year). When comparing the calculated and actual values of precipitation, it was revealed that the maximum error was observed in 2002 and amounted to 95 mm, or 15.8%.

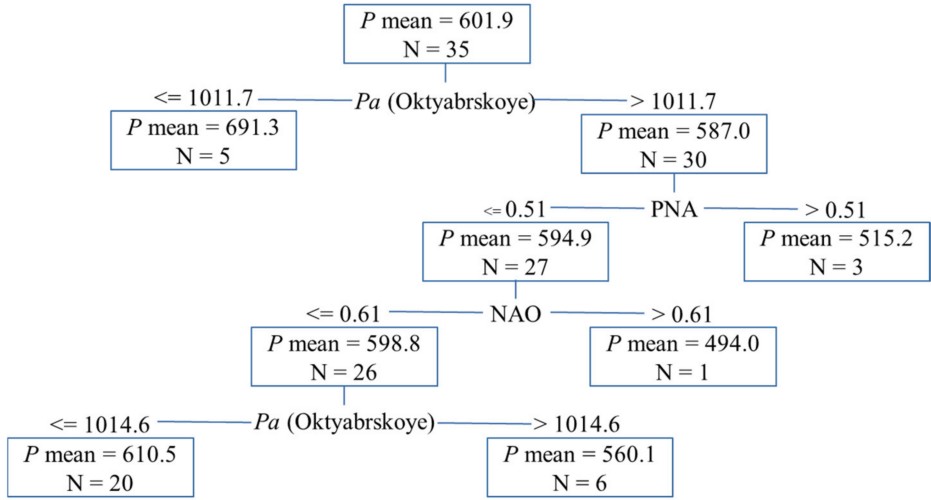

**Figure 5.** Decision tree for annual values of precipitation (mm/year) of the Ob's basin, consisting of 4 branches for 1981–2015.

## 4. Conclusions

The study of variability of moistening in the territory of Siberian Russia is a very comprehensive task due to almost complete absence of stationary meteorological stations in most of Siberia. This work involved only 32 stations located mainly in the valleys of large rivers. As a result of calculations, the estimates of potential evapotranspiration, precipitation and potential evapotranspiration coefficient have been obtained in the period 1981–2015. A complex delimitation of the territory of Siberia has been carried out according to the nature of interannual fluctuations in the annual values of characteristics of moistening. Four regions are shown to be identified, three of which being located in the basins of the largest rivers: the Ob, the Yenisei, the Lena. The fourth region—Baikalsky, represents the area around Lake Baikal.

The interannual variability of $E_0$ is almost equal in all four regions. The coefficient of variation amounts to 7–8%. The interannual variability of precipitation is significantly higher, its coefficient of variation varying from 11% (the Ob basin) to 16% (the Baikal region). The variability of $E_0/P$ is even higher, reaching 23% in the Baikal region

Trend analysis has shown the potential evapotranspiration in Siberia to be growing, but the trend is significant only in the Ob basin and the Baikal region. Precipitation, excluding the Baikal region, is also increasing. Significant trends are observed in the Yenisei and Lena basins. As for the potential evapotranspiration, a significant trend takes place only in the Baikal region due to the rapid growth of $E_0$.

The modeling of the annual values of characteristics of moistening for the selected quasi-homogeneous regions has been carried out using the decision trees method with the CART algorithm. In order to compare the obtained simulation results with each other, a variant of a tree with 4 branches has been considered to be the optimal model. For all models the coefficient of determination $R^2$ has shown to be almost equal and describe about

two-thirds of the variance of the original variable (0.57–0.73). In the models of annual values of potential evapotranspiration, the main predictor, as expected, is the air temperature at the station of the region under consideration. In precipitation models, the contribution of local and external circulation factors to interannual precipitation fluctuations is equal.

**Author Contributions:** V.M. conceived of the presented idea and developed the theory. S.G. and J.M. performed the calculations and analysed the data. V.M. wrote the manuscript with input from all authors. All authors have read and agreed to the published version of the manuscript.

**Funding:** The work has been carried out within the framework of the Russian Government Contract FSZU2020-0009 "Study of physical, chemical and biological processes in the atmosphere and hydrosphere under conditions of climate change and anthropogenic impacts".

**Institutional Review Board Statement:** Not applicable.

**Informed Consent Statement:** Not applicable.

**Data Availability Statement:** Publicly available datasets were analyzed in this study. These data can be found here: http://aisori.meteo.ru/ClimateR, password access; https://cds.climate.copernicus.eu/cdsapp#!/dataset/10.24381/cds.68d2bb30? tab = overview, password access; https://www.cpc.ncep.noaa.gov/data/teledoc/telecontents.shtml, free access, all accessed on 1 July 2021.

**Conflicts of Interest:** The authors declare no conflict of interest.

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
