# Peer review of "Assessment of Interannual Variability of Moistening of Siberian Territory According to Observational Data"

_water, doi:10.3390/w13162200_

Round 1

Reviewer 1 Report

Line 9. “……The RIHMI-WDC archive” add an explanation of the abbreviation at the first time given in the text.

Line 12. “…method of M.I. Budyko has been used” the same as previous.

Line 40-45. Some appropriate references must be added as below:

  1. Stefanidis, S., & Stathis, D. (2018). Spatial and temporal rainfall variability over the Mountainous Central Pindus (Greece). Climate, 6(3), 75.
  2. Tashilova, A. A., Ashabokov, B. A., Kesheva, L. A., & Teunova, N. V. (2019). Analysis of climate change in the caucasus region: end of the 20th–beginning of the 21st century. Climate, 7(1), 11.
  3. Marshall, G. J. (2021). Decadal variability in the impact of atmospheric circulation patterns on the winter climate of northern Russia. Journal of Climate, 34(3), 1005-1021.

Line 58-64. This paragraph must be re-written. Some more information of the evaporation must be given. Add comments about the different methods as the Thornthwaite method is the easiest way of calculating PET in data-scared areas. However, PET is preferably estimated, in terms of accuracy, by using the Penman formula.

In the last paragraph of the introduction the novelty of the current research must be clearly identified and the scientific questions of this paper given in detail.

Are the meteorological data check for homogeneity. This is crucial and necessary before a climate study. It needs to add.

Line 242.

Separate the chapters. The discussion must be a separate chapter and discuss in depth the results. Also, comparison with other similar studies must be investigated.

Finally, add some proposals for future research.  

Reviewer 2 Report

... please, see the attached file. Thank you.

Round 2

Reviewer 1 Report

Despite in their response author refer that address some comment in the text we do not found the appropriate references as bellow. Please check and revised.

  1. Stefanidis, S., & Stathis, D. (2018). Spatial and temporal rainfall variability over the Mountainous Central Pindus (Greece). Climate, 6(3), 75.
  2. Tashilova, A. A., Ashabokov, B. A., Kesheva, L. A., & Teunova, N. V. (2019). Analysis of climate change in the caucasus region: end of the 20th–beginning of the 21st century. Climate, 7(1), 11.
  3. Marshall, G. J. (2021). Decadal variability in the impact of atmospheric circulation patterns on the winter climate of northern Russia. Journal of Climate, 34(3), 1005-1021.

Reviewer 2 Report

The work has been improved, as a whole, although some shortcomings remain.

If the image from the 350 line is Figure 1., in this case the image from the line 106 represents Figure 0. ?
Please, clarify, also, where you cited the figures, in the text body.

If the authors made the figure from 106 line, which is the spatial data source, that must be cited? 
If the authors not made this figure, which is it source?
Are the black lines (limits) of Siberian provinces cartographically real, or these are approximately vectorised by the authors? Its directions are so bizarre ...

Where is the legend from the figure placed at the 427 line?
In the same figure, you placed a 1st degree equation: if E0=0.97*t+426.6, please, mathematically calculate for us the year parameter for the 400mm value of E0 ... :)) Be realist and remove the equation, when you have the time on X axis ... R2 can remain ...

Thank you.
